# Close-kin mark-recapture methods to estimate demographic parameters of mosquitoes

**Yogita Sharma**[1,2☯], **Jared B. Bennett**[3☯], **Gordana Rašić**[4], **John M. Marshall**[1,5]*

**1** Divisions of Biostatistics and Epidemiology, School of Public Health, University of California, Berkeley, California, United States of America, **2** Department of Mathematics and Statistics, University of Victoria, Victoria, British Columbia, Canada, **3** Biophysics Graduate Group, Division of Biological Sciences, College of Letters and Science, University of California, Berkeley, California, United States of America, **4** Mosquito Genomics, QIMR Berghofer Medical Research Institute, Brisbane, Queensland, Australia, **5** Innovative Genomics Institute, University of California, Berkeley, California, United States of America

☯ These authors contributed equally to this work.
* john.marshall@berkeley.edu

**Data Availability Statement:** The source code for the individual-based mosquito simulation model is available at https://github.com/GilChrist19/mPlex, with a tagged version corresponding to submission

## Abstract

Close-kin mark-recapture (CKMR) methods have recently been used to infer demographic parameters such as census population size and survival for fish of interest to fisheries and conservation. These methods have advantages over traditional mark-recapture methods as the mark is genetic, removing the need for physical marking and recapturing that may interfere with parameter estimation. For mosquitoes, the spatial distribution of close-kin pairs has been used to estimate mean dispersal distance, of relevance to vector-borne disease transmission and novel biocontrol strategies. Here, we extend CKMR methods to the life history of mosquitoes and comparable insects. We derive kinship probabilities for mother-offspring, father-offspring, full-sibling and half-sibling pairs, where an individual in each pair may be a larva, pupa or adult. A pseudo-likelihood approach is used to combine the marginal probabilities of all kinship pairs. To test the effectiveness of this approach at estimating mosquito demographic parameters, we develop an individual-based model of mosquito life history incorporating egg, larva, pupa and adult life stages. The simulation labels each individual with a unique identification number, enabling close-kin relationships to be inferred for sampled individuals. Using the dengue vector *Aedes aegypti* as a case study, we find the CKMR approach provides unbiased estimates of adult census population size, adult and larval mortality rates, and larval life stage duration for logistically feasible sampling schemes. Considering a simulated population of 3,000 adult mosquitoes, estimation of adult parameters is accurate when ca. 40 adult females are sampled biweekly over a three month period. Estimation of larval parameters is accurate when adult sampling is supplemented with ca. 120 larvae sampled biweekly over the same period. The methods are also effective at detecting intervention-induced increases in adult mortality and decreases in population size. As the cost of genome sequencing declines, CKMR holds great promise for characterizing the demography of mosquitoes and comparable insects of epidemiological and agricultural significance.

of this paper available at https://github.com/
GilChrist19/mPlex/releases/tag/0.1.1.
Documentation, including for installation and
dependencies, is available at https://github.com/
GilChrist19/mPlex/tree/master/CKMR. The source
code for running simulations and inferring
parameters based on the pseudo-likelihood of the
kinship data is available at https://github.com/
MarshallLab/CKMR. Both sets of code are available
under the GPL3 License and are free for other
groups to modify and extend as needed.

**Funding:** YS, GR and JMM were supported by a
National Institutes of Health R01 Grant
(1R01AI143698-01A1) awarded to JMM and GR.
YS, JBB, GR and JMM were supported by a
DARPA Safe Genes Program Grant (HR0011-17-2-
0047) awarded to JMM. The funders had no role in
the study design, data collection and analysis,
decision to publish, or preparation of the
manuscript.

**Competing interests:** The authors have declared
that no competing interests exist.

## Author summary

Close-kin mark-recapture (CKMR) methods are a genetic analogue of traditional mark-recapture methods in which the frequency of marked individuals in a sample is used to infer demographic parameters such as census population size and mean dispersal distance. In CKMR, the mark is a close-kin relationship between individuals (parents and offspring, siblings, etc.). While CKMR methods have mostly been applied to aquatic species to date, opportunities exist to apply them to insects and other terrestrial species. Here, we explore the application of CKMR to mosquitoes, with *Aedes aegypti*, a primary vector of dengue, chikungunya and yellow fever, as a case study. By analyzing simulated *Ae. aegypti* populations, we find the CKMR approach provides unbiased estimates of adult census population size, adult and larval mortality rates, and larval life stage duration, and may be informative of intervention impact. Optimal sampling schemes are compatible with *Ae. aegypti* ecology and field studies. This study represents the first theoretical exploration of the application of CKMR to an insect species, and demonstrates its potential for characterizing the demography of insects of epidemiological and agricultural importance.

## 1 Introduction

In the last few years, there has been a growth of interest in close-kin mark-recapture (CKMR) methods to characterize the demography of wild populations [1]. These methods are analogous to traditional mark-recapture methods, which estimate census population size and other demographic parameters based on the recapture rates of marked individuals. The advantages of CKMR methods stem from the mark being a genetically-inferred close-kin relationship, removing the need for physical marking and recapturing. Initial applications of these methods have included a wide range of fish species—southern bluefin tuna [2], white sharks [3], brook trout [4] and Atlantic salmon [5]. Fish provide a good case for CKMR because their populations are well-mixing, physical marking and recapturing pose logistical challenges, and there is a willingness to invest in population size estimates given their importance to fisheries and conservation [1]. CKMR studies on fish have also estimated annual juvenile and adult survival probabilities and rates of population growth [2, 3].

As high-throughput genomic sequencing, which enables accurate kinship estimation, becomes cheaper, it is expected that CKMR methods will be applied to an increasing number of species. For insects, two recent studies used the spatial distribution of close-kin pairs to characterize dispersal patterns of *Aedes aegypti* [6, 7], the mosquito vector of dengue, Zika, chikungunya and yellow fever. Both studies were set in urban landscapes—in Malaysia [6] and Singapore [7]—where mosquitoes inhabit high-rise apartment buildings. These locations were chosen to support releases of *Wolbachia*-infected mosquitoes intended for population replacement [8] and suppression [9]. Characterizing mosquito movement is important to understanding the spatial transmission of vector-borne diseases [10], and to designing optimal biocontrol strategies, such as those involving *Wolbachia*, for vector-borne disease control. By analyzing close-kin pairs, these two studies estimated mean dispersal distances in agreement with previous mark-recapture studies [7, 11], and isolated a radius of dispersal specific to *Ae. aegypti* oviposition behavior [6].

In this paper, we extend the CKMR formalism described by Bravington *et al.* [1] to mosquitoes, using *Ae. aegypti* as a case study, in order to derive demographic parameters from close-kin pairs. These methods involve deriving "kinship probabilities" describing the chance that a given individual is related to another in the population. These are calculated as the

reproductive output having a given kinship relationship divided by the total reproductive output of all adult females in the population, and depend upon a parameterized model of life history and mating behavior, including egg production and mortality rates. Because the age of adult *Ae. aegypti* mosquitoes is difficult to estimate in the field, age must be accommodated as a latent variable, with marginal kinship probabilities being calculated by considering all consistent event histories. For fish species to which CKMR methods have been applied thus far, full-siblings are rare as adults tend to be polygamous [3]. In contrast, for mosquitoes, full-siblings are common as adult females tend to mate only once, soon after emergence, and lay eggs from this mating event over an extended period. Mosquito half-siblings are also common, and tend to be paternal (i.e., have the same father and different mothers). Taking these considerations into account, we derive kinship probabilities for mother-offspring, father-offspring, full-sibling and half-sibling pairs where either individual in each pair may be a larva, pupa or adult. A pseudo-likelihood approach is used to combine the marginal probabilities of all kinship pairs [1].

To test the effectiveness of this approach at estimating mosquito demographic parameters, we develop an individual-based model of mosquito life history, incorporating egg, larva, pupa and adult life stages. By labeling each individual with a unique identification number (IN) and tracking parental INs, this enables close-kin relationships to be inferred for sampled individuals. As studies of aquatic species have shown, a parsimonious individual-based simulation of life history allows a variety of CKMR sampling schemes to be explored, and for effectiveness at parameter estimation to be assessed [12, 13]. The short generation time of mosquitoes—less than a month for *Ae. aegypti* [14]—means that sampling may take place over a few months, as opposed to several years for long-lived fish species [2]. Open questions regarding sampling schemes for mosquitoes relate to the required sample size, optimal frequency (e.g., daily, biweekly or weekly), duration (i.e., number of months), and distribution of collections across larval, pupal and adult life stages in order to estimate population size, mortality rates, and durations of juvenile life stages. Here, we use our simulation model and CKMR framework to address these questions, and in doing so, provide a case study for CKMR applications to comparable insects of epidemiological and agricultural significance.

## 2 Materials and methods

### 2.1 Mosquito population dynamics

We use a discrete-time version of the lumped age-class model [15, 16], applied to mosquitoes [17], as the basis for our population simulation and CKMR analysis (Fig 1). This model considers discrete life history stages—egg (E), larva (L), pupa (P) and adult (A)—with sub-adult stages having defined durations—$T_E$, $T_L$ and $T_P$ for eggs, larvae and pupae, respectively. We use a daily time-step, since mosquito samples tend to be recorded by day, and this is adequate to model the organism's population dynamics [18]. Daily mortality rates vary according to life stage—$\mu_E$, $\mu_L$, $\mu_P$ and $\mu_A$ for eggs, larvae, pupae and adults, respectively—and density-dependent mortality occurs at the larval stage. Mortality rates are assumed to be independent of age. The sex of an emergent pupa is drawn from a Bernoulli distribution with probability 0.5 such that, on average, half of emerging adults are female (F) and half are male (M). Females mate once upon emergence, and retain the genetic material from that mating event for the remainder of their lives. Males mate at a rate equal to the female emergence rate which, for a population at equilibrium, is equal to the female mortality rate, $\mu_A$. Females lay eggs at a rate, $\beta$, which is assumed to be independent of age.

Default life history and demographic parameters for *Ae. aegypti* are listed in Table 1. Given the difficulty of measuring juvenile stage mortality rates in the wild, these are chosen for

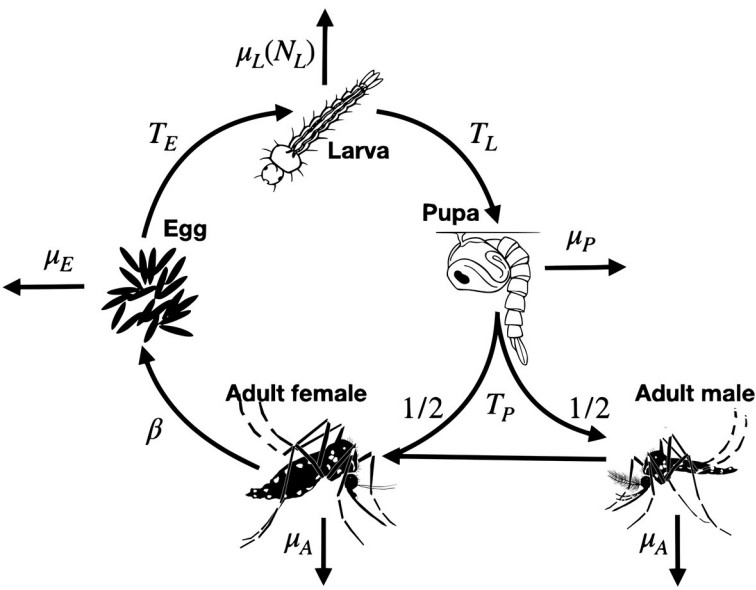

**Fig 1. The lumped age-class model of mosquito life history.** Mosquitoes are divided into four life stages: egg, larva, pupa and adult. The durations of the sub-adult stages are $T_E$, $T_L$ and $T_P$ for eggs, larvae and pupae, respectively. Sex is modeled at the adult stage, with half of pupae developing into females and half developing into males. Daily mortality rates vary by life stage—$\mu_E$, $\mu_L$, $\mu_P$ and $\mu_A$ for eggs, larvae, pupae and adults, respectively. Density-dependent mortality occurs at the larval stage and is a function of the total number of larvae, $N_L$. Females mate once upon emergence, and retain the genetic material from that mating event for the remainder of their lives. Males mate at a rate equal to the female emergence rate. Females lay eggs at a rate, $\beta$.

consistency with observed population growth rates in the absence of density-dependence (see S1 Text §1.1 for formulae and derivations). Larval mortality increases with larval density and, according to the lumped age-class model, reaches a set value when the population is at equilibrium. Although mosquito populations vary seasonally, we assume a constant adult population size, $N_A$, for this CKMR analysis, and restrict sampling to a maximum period of four months, corresponding to a season. Minor population size fluctuations occur in the simulation model due to sampling and stochasticity.

## 2.2 Kinship probabilities

Following the methodology of Bravington *et al.* [1], we now derive kinship probabilities for mother-offspring, father-offspring, full-sibling and half-sibling pairs based on the lumped age-

**Table 1. Demographic and life history parameters for *Aedes aegypti* mosquitoes.**

| Parameter: | Definition: | Value: | References: |
|---|---|---|---|
| $N_A$ | Adult population size | 3000 | [19–21] |
| $\mu_A$ | Adult mortality rate | 0.09 / day | [22] |
| $\beta$ | Female fecundity | 20 / day | [23] |
| $T_E$ | Duration of egg stage | 2 days | [14] |
| $T_L$ | Duration of larval stage | 5 days | [14] |
| $T_P$ | Duration of pupal stage | 1 day | [14] |
| $\mu_E$ | Egg mortality rate | 0.175 / day | S1 Text §1, [24] |
| $\mu_L$ | Larval mortality rate | 0.554 / day | S1 Text §1 |
| $\mu_P$ | Pupal mortality rate | 0.175 / day | S1 Text §1, [24] |

**Table 2. Kinship categories, sampled life stages, sampling times, and probability symbols used in close-kin mark-recapture analysis.**

| Kinship category: | Sampled life stages: | Probability symbol: | Equations: |
|---|---|---|---|
| Mother-offspring | Adult female ($t_1$), larva ($t_2$) | $P_{MOL}(t_2\|t_1)$ | §2.2.1 |
| | Adult female ($t_1$), adult ($t_2$) | $P_{MOA}(t_2\|t_1)$ | S1 Text §2.1 |
| | Adult female ($t_1$), pupa ($t_2$) | $P_{MOP}(t_2\|t_1)$ | S1 Text §2.1 |
| Father-offspring | Adult male ($t_1$), larva ($t_2$) | $P_{FOL}(t_2\|t_1)$ | S1 Text §2.2 |
| | Adult male ($t_1$), adult ($t_2$) | $P_{FOA}(t_2\|t_1)$ | §2.2.2 |
| | Adult male ($t_1$), pupa ($t_2$) | $P_{FOP}(t_2\|t_1))$ | S1 Text §2.2 |
| Full-siblings | Larva ($t_1$), larva ($t_2$) | $P_{FSLL}(t_2\|t_1)$ | §2.2.3 |
| | Adult ($t_1$), adult ($t_2$) | $P_{FSAA}(t_2\|t_1)$ | S1 Text §2.3 |
| | Larva ($t_1$), adult ($t_2$) | $P_{FSLA}(t_2\|t_1)$ | S1 Text §2.3 |
| | Adult ($t_1$), larva ($t_2$) | $P_{FSAL}(t_2\|t_1)$ | S1 Text §2.3 |
| | Pupa ($t_1$), pupa ($t_2$) | $P_{FSPP}(t_2\|t_1)$ | S1 Text §2.3 |
| | Pupa ($t_1$), larva ($t_2$) | $P_{FSPL}(t_2\|t_1)$ | S1 Text §2.3 |
| | Larva ($t_1$), pupa ($t_2$) | $P_{FSLP}(t_2\|t_1)$ | S1 Text §2.3 |
| | Pupa ($t_1$), adult ($t_2$) | $P_{FSPA}(t_2\|t_1)$ | S1 Text §2.3 |
| | Adult ($t_1$), pupa ($t_2$) | $P_{FSAP}(t_2\|t_1)$ | S1 Text §2.3 |
| Half-siblings | Larva ($t_1$), larva ($t_2$) | $P_{HSLL}(t_2\|t_1)$ | §2.2.4 |
| | Adult ($t_1$), adult ($t_2$) | $P_{HSAA}(t_2\|t_1)$ | S1 Text §2.4 |
| | Larva ($t_1$), adult ($t_2$) | $P_{HSLA}(t_2\|t_1)$ | S1 Text §2.4 |
| | Adult ($t_1$), larva ($t_2$) | $P_{HSAL}(t_2\|t_1)$ | S1 Text §2.4 |
| | Pupa ($t_1$), pupa ($t_2$) | $P_{HSPP}(t_2\|t_1)$ | S1 Text §2.4 |
| | Pupa ($t_1$), larva ($t_2$) | $P_{HSPL}(t_2\|t_1)$ | S1 Text §2.4 |
| | Larva ($t_1$), pupa ($t_2$) | $P_{HSLP}(t_2\|t_1)$ | S1 Text §2.4 |
| | Pupa ($t_1$), adult ($t_2$) | $P_{HSPA}(t_2\|t_1)$ | S1 Text §2.4 |
| | Adult ($t_1$), pupa ($t_2$) | $P_{HSAP}(t_2\|t_1)$ | S1 Text §2.4 |

class mosquito life history model. Each kinship probability is calculated as the reproductive output having that relationship divided by the total reproductive output of all adult females in the population. In each case, we consider two individuals (adult, larva or pupa) sampled at known times, $t_1$ and $t_2$, with probability symbols and references to equations listed in Table 2. Important details to note for this analysis are that: i) mosquito sampling is lethal, ii) although age is a latent variable, temporal information is captured in the life stages of sampled individuals, and iii) mosquito mating behaviour, in which females mate once upon emergence and males mate throughout their adult lifespan, is reflected in the calculations.

**2.2.1 Mother-offspring.** Let us begin with the simplest possible kinship probability, $P_{MOL}(t_2|t_1)$, which represents the probability that, given an adult female sampled on day $t_1$, a larva sampled on day $t_2$ is her offspring. This can be expressed as the relative larval reproductive output on day $t_2$ of an adult female sampled on day $t_1$:

$$P_{MOL}(t_2|t_1) = \frac{\mathbb{E}[\text{Larval offspring at time } t_2 \text{ from an adult female sampled at time } t_1]}{\mathbb{E}[\text{Larval offspring at time } t_2 \text{ from all adult females}]} = \frac{E_{MOL}(t_2|t_1)}{E_L}. \quad (1)$$

Here, $E_{MOL}(t_2|t_1)$ represents the expected number of surviving larval offspring on day $t_2$ from an adult female sampled on day $t_1$, and $E_L$ represents the expected number of surviving larval offspring from all adult females in the population at times consistent with the time of larval sampling. Note that, since we are assuming a constant population size, $E_L$ is independent of

time and is given by:

$$E_L = \sum_{y_2=0-T_E-(T_L-1)}^{0-T_E} N_F \times \beta \times (1-\mu_E)^{T_E} \times (1-\mu_L)^{(0-y_2-T_E)}. \tag{2}$$

Here, $N_F$ represents the equilibrium adult female population size (which is equal to half the equilibrium adult population size, $N_A/2$), and $y_2$ represents the day of egg laying. Considering day 0 as the reference day (in place of $t_2$), the egg must have been laid between days $(0 - T_E - (T_L - 1))$ and $(0 - T_E)$ (Fig 2A). Eq 2 represents the expected number of offspring laid by all adult females in the population that survive the egg and larva stages up to the time of sampling (day 0) and is graphically depicted in Fig 2A.

$E_{MOL}(t_2|t_1)$, on the other hand, is specific to the sampled adult female and the day of larval sampling, $t_2$. This is graphically depicted in Fig 2B, and is given by:

$$E_{MOL}(t_2|t_1) = \sum_{y_2=t_2-T_E-(T_L-1)}^{t_2-T_E} (1-\mu_A)^{(t_1-y_2)} \times \left( \mathbb{I}[(t_1-T_A) < y_2 \leq t_1] \times \beta \times (1-\mu_E)^{T_E} \times (1-\mu_L)^{(t_2-y_2-T_E)} \right). \tag{3}$$

Here, the day of egg-laying, $y_2$, is summed over days $(t_2 - T_E - (T_L - 1))$ through $(t_2 - T_E)$, for consistency with the larva being present on the day of sampling (Fig 2B). The first term in the summation represents the probability that the adult female sampled on day $t_1$ is alive on the day of egg-laying, and the second term (in larger brackets) represents the expected surviving larval output of this adult female on day $t_2$. This latter term is equal to their daily egg production, $\beta$, multiplied by the proportion of eggs that survive the egg and larva stages from the day they were laid up to the day of sampling. An indicator function is included to limit consideration to cases where the day of egg-laying lies within the adult female's possible lifetime—i.e., between days $t_1$ and $(t_1 - T_A)$, where $T_A$ represents the maximum possible age of an adult mosquito. Although adult lifetime is exponentially-distributed, a value of $T_A$ may be chosen that captures most of this distribution and leads to accurate parameter inference.

Extending the mother-offspring kinship probability for pupal and adult offspring is straightforward. These extensions are provided in S1 Text §2.1, and are also described for the father-adult offspring case below.

**2.2.2 Father-offspring.**   Next, we consider the father-adult offspring kinship probability, $P_{FOA}(t_2|t_1)$, which represents the probability that, given an adult male sampled on day $t_1$, an adult sampled on day $t_2$ is his offspring. This can be expressed as the relative adult reproductive output on day $t_2$ of adult females that mated with an adult male sampled on day $t_1$:

$$P_{FOA}(t_2|t_1) = \frac{\mathbb{E}[\text{Adult offspring at time } t_2 \text{ from an adult male sampled at time } t_1]}{\mathbb{E}[\text{Adult offspring at time } t_2 \text{ from all adult females}]} = \frac{E_{FOA}(t_2|t_1)}{E_A}. \tag{4}$$

Here, $E_{FOA}(t_2|t_1)$ represents the expected number of surviving adult offspring on day $t_2$ of an adult male sampled on day $t_1$, and $E_A$ represents the expected number of surviving adult offspring from all adult females at times consistent with the time of adult offspring sampling. Assuming a population at equilibrium, $E_A$ is independent of time and is given by:

$$E_A = \sum_{y_2=0-T_E-T_L-T_P-(T_A-1)}^{0-T_E-T_L-T_P} N_F \times \beta \times (1-\mu_E)^{T_E} \times (1-\mu_L)^{T_L} \times (1-\mu_P)^{T_P} \times (1-\mu_A)^{(0-y_2-T_E-T_L-T_P)}. \tag{5}$$

Here, considering day 0 as the reference day (in place of $t_2$), the day of egg-laying, $y_2$, is summed over days $(0 - T_E - T_L - T_P - (T_A - 1))$ through $(0 - T_E - T_L - T_P)$, for consistency with the adult offspring being present on the day of sampling (Fig 2C). Eq 5 therefore

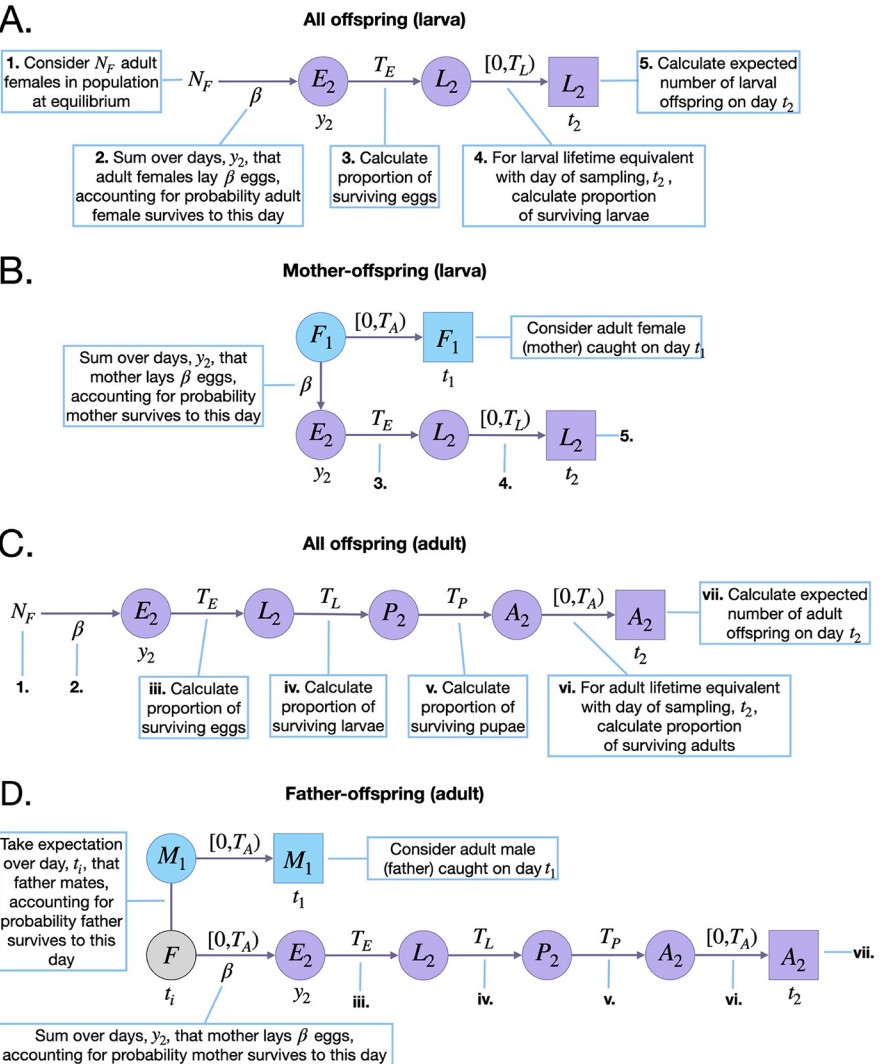

**Fig 2. Schematic representation of parent-offspring kinship probabilities.** Parameters and state variables are as defined in Table 1 and §2.1. Subscript 1 refers to the parent (blue), and subscript 2 refers to the offspring (purple, the perspective from which probabilities are calculated). Circles represent living individuals and squares represent sampled individuals. Parents are sampled on day $t_1$, eggs are laid on day $y_2$, and offspring are sampled on day $t_2$. Offspring kinship probabilities are the ratio of the expected number of surviving offspring from a given adult on day $t_2$, and the expected number of surviving offspring from all adult females on this day. The expected number of surviving offspring from all adult females requires considering days of egg-laying consistent with larval ages at sampling in the range $[0, T_L)$ (for larval offspring) **(A)**, or adult ages at sampling in the range $[0, T_A)$ (for adult offspring) **(C)**. Calculating the expected number of surviving larval offspring on day $t_2$ from an adult female requires considering days of egg-laying, $y_2$, consistent with maternal ages at egg-laying in the range $[0, T_A)$, and with larval offspring ages at sampling in the range $[0, T_L)$ **(B)**. For father-adult offspring pairs, this requires considering days of mating, $t_i$, and egg-laying, $y_2$, consistent with maternal ages at egg-laying, and paternal and adult offspring ages at sampling in the range $[0, T_A)$ **(D)**.

represents the expected number of offspring laid by all adult females in the population that survive the egg, larva, pupa and adult stages up to the time of sampling (day 0) and is graphically depicted in Fig 2C.

$E_{FOA}(t_2|t_1)$ is graphically depicted in Fig 2D. Each adult female mates once upon emergence and, since there are equal numbers of adult females and males in the population, each adult male mates on average once in their lifetime too. The day of this mating event, $t_i$, is unknown

and so, in calculating $E_{FOA}(t_2|t_1)$, we treat this as a latent variable and take an expectation over all possible values it can take:

$$E_{FOA}(t_2|t_1) = \sum_{t_i=t_1-(T_A-1)}^{t_1} p_A(t_1 - t_i) \times E_{FOA}(t_2|t_1, t_i). \tag{6}$$

Here, the expectation over the day of mating, $t_i$, is taken over days $(t_1 - (T_A - 1))$ through $t_1$, for consistency with the day of adult male sampling (Fig 2D). The term $E_{FOA}(t_2|t_1, t_i)$ represents the expected number of adult offspring on day $t_2$, conditional upon the adult male being sampled on day $t_1$ and the day of mating being $t_i$, and $p_A(t)$ represents the probability that a given adult in the population has age $t$. Here, the probability that an adult has age $(t_1 - t_i)$ is equivalent to the probability that the mating event occurred on day $t_i$. In general, $p_A(t)$ follows from the daily adult survival probability, $(1 - \mu_A)$, and is given by:

$$p_A(t) = (1 - \mu_A)^t \Big/ \sum_{t_j=0}^{T_A-1} (1 - \mu_A)^{t_j}. \tag{7}$$

$E_{FOA}(t_2|t_1, t_i)$ is then given by:

$$E_{FOA}(t_2|t_1, t_i) = \sum_{y_2=t_i}^{t_i+(T_A-1)} (1 - \mu_A)^{(y_2-t_i)} \times \left( \begin{array}{c} \mathbb{I}[(y_2 + T_E + T_L + T_P) \leq t_2 < (y_2 + T_E + T_L + T_P + T_A)] \\ \times \beta \times (1 - \mu_E)^{T_E} \times (1 - \mu_L)^{T_L} \\ \times (1 - \mu_P)^{T_P} \times (1 - \mu_A)^{(t_2-y_2-T_E-T_L-T_P)} \end{array} \right). \tag{8}$$

Here, the day of egg-laying, $y_2$, is summed over days $t_i$ through $(t_i + (T_A - 1))$, for consistency with the mother's potential lifespan (Fig 2D). The first term in the summation represents the probability that the mother is alive on the day of egg-laying, and the second term (in larger brackets) represents the expected surviving adult output of this adult female on day $t_2$. This latter term is equal to their daily egg production, $\beta$, multiplied by the proportion of eggs that survive the egg, larva, pupa and adult stages from the day they were laud up to the day of sampling. An indicator function is included to limit consideration to cases where the day of adult offspring sampling, $t_2$, lies within their possible adult lifetime—i.e., between days $(y_2 + T_E + T_L + T_P)$ and $(y_2 + T_E + T_L + T_P + T_A)$.

Extending the father-offspring kinship probability for larval and pupal offspring is straightforward, involving similar adaptations as per this case. These extensions are provided in S1 Text §2.2.

**2.2.3 Full-siblings.** Next, we consider the full-sibling kinship probability for larva-larva pairs, $P_{FSLL}(t_2|t_1)$, which represents the probability that, given a larva sampled on day $t_1$, a larva sampled on day $t_2$ is their full-sibling. This can be expressed as the relative larval reproductive output on day $t_2$ of the mother of a larva sampled on day $t_1$:

$$P_{FSLL}(t_2|t_1) = \frac{\mathbb{E}[\text{Larvae at time } t_2 \text{ that are full-siblings of a larva sampled at time } t_1]}{\mathbb{E}[\text{Larval offspring at time } t_2 \text{ from all adult females}]} = \frac{E_{FSLL}(t_2|t_1)}{E_L}. \tag{9}$$

Here, $E_{FSLL}(t_2|t_1)$ represents the expected number of surviving larvae on day $t_2$ that are full-siblings of a larva sampled on day $t_1$ and is graphically depicted in Fig 3A. $E_L$ is given by Eq 2. For convenience, let us refer to the larva sampled on day $t_1$ as individual 1. To calculate

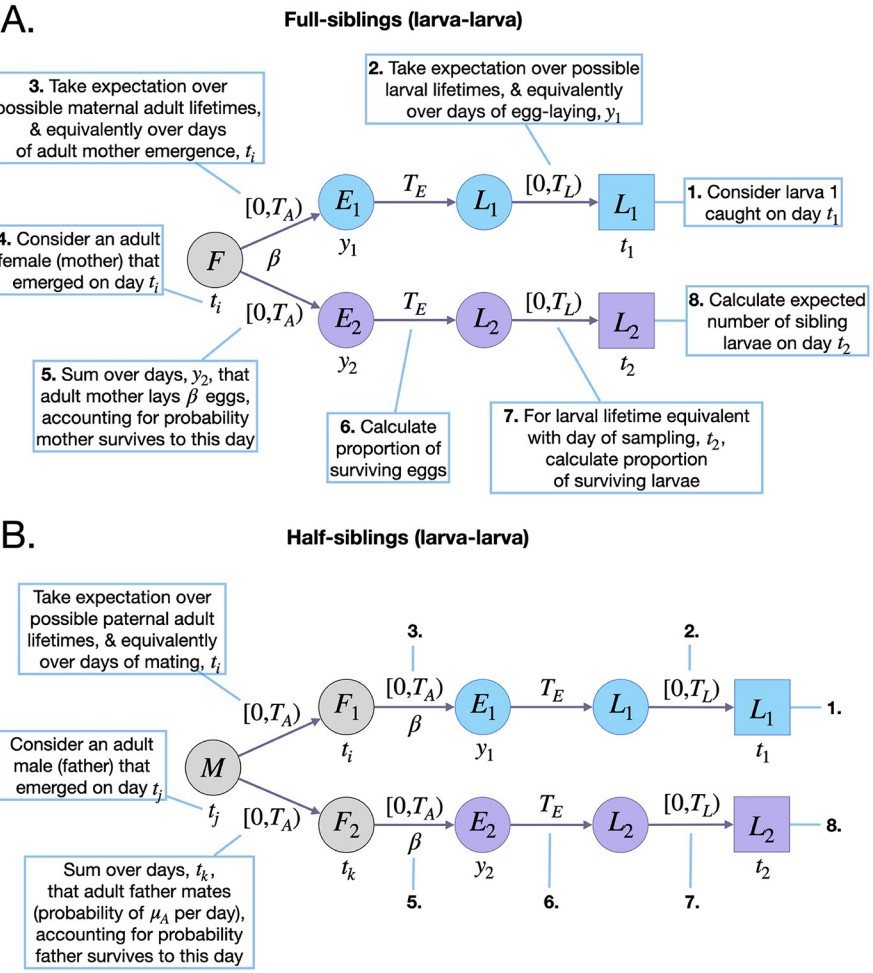

**Fig 3. Schematic representation of sibling kinship probabilities.** Parameters and state variables are as defined in Table 1 and §2.1. Subscript 1 refers to the reference sibling (blue), and subscript 2 refers to the sibling from whose perspective the probabilities are calculated (purple). Circles represent living individuals and squares represent sampled individuals. The reference sibling is sampled on day $t_1$ and laid on day $y_1$. Sibling 2 is sampled on day $t_2$ and laid on day $y_2$. Sibling kinship probabilities are the ratio of the expected number of surviving siblings of a given individual on day $t_2$, and the expected number of surviving offspring from all adult females on this day. Calculating the expected number of surviving larval full-siblings of a larva requires considering days of their mother emerging as an adult, $t_i$, and of egg-laying, $y_1$ and $y_2$, that are consistent with maternal ages at egg-laying in the range $[0, T_A)$, and with larval ages at sampling in the range $[0, T_L)$ **(A)**. Calculating the expected number of surviving larval half-siblings of a larva requires considering days of their father emerging as an adult, $t_j$, their mothers emerging as adults, $t_i$ and $t_k$, and of egg-laying, $y_1$ and $y_2$, that are consistent with paternal ages at mating in the range $[0, T_A)$, maternal ages at egg-laying in the range $[0, T_A)$, and larval ages at sampling in the range $[0, T_A)$ **(B)**.

$E_{FSLL}(t_2|t_1)$, there are two unknown event times that we treat as latent variables and take an expectation over—i) the day that egg 1 is laid, $y_1$, and ii) the day that individual 1's mother emerges as an adult, $t_i$:

$$E_{FSLL}(t_2|t_1) = \sum_{y_1=t_1-T_E-(T_L-1)}^{t_1-T_E} p_L(t_1 - y_1 - T_E) \times \sum_{t_i=y_1-(T_A-1)}^{y_1} p_A(y_1 - t_i) \times E_{FSLL}(t_2|t_1, y_1, t_i). \quad (10)$$

Here, the expectation over the day that egg 1 is laid, $y_1$, is taken over days $(t_1 - T_E - (T_L - 1))$ through $(t_1 - T_E)$, for consistency with the day that larva 1 is sampled, and the expectation over the day of their mother's emergence, $t_i$, is taken over days $(y_1 - (T_A - 1))$ through $y_1$, so that egg 1 may be laid during their mother's potential lifetime (Fig 3A). The term $E_{FSLL}(t_2|t_1, y_1, t_i)$ represents the expected number of surviving larvae on day $t_2$ that are full-siblings of larva 1, conditional upon egg 1 being laid on day $y_1$, and their mother emerging as an adult on day $t_i$. $p_A(y_1 - t_i)$ represents the probability that their mother has age $(y_1 - t_i)$, which is equivalent to the probability that their mother emerged on day $t_i$. Additionally, $p_L(t)$ represents the probability that a given larva in the population has age $t$, and the probability that larva 1 has age $(t_1 - y_1 - T_E)$ is equivalent to the probability that the egg was laid on day $y_1$. In general, $p_A(t)$ is given by Eq 7, and $p_L(t)$ follows from the daily larval survival probability, $(1 - \mu_L)$, and is given by:

$$p_L(t) = (1 - \mu_L)^t / \sum_{t_j=0}^{T_L-1} (1 - \mu_L)^{t_j}. \tag{11}$$

$E_{FSLL}(t_2|t_1, y_1, t_i)$ is then given by:

$$E_{FSLL}(t_2|t_1, y_1, t_i) = \sum_{y_2=t_i}^{t_i+(T_A-1)} (1 - \mu_A)^{(y_2-t_i)} \times \begin{pmatrix} \mathbb{I}[(t_2 - T_E - T_L) < y_2 \leq (t_2 - T_E)] \\ \times \beta \times (1 - \mu_E)^{T_E} \times (1 - \mu_L)^{(t_2-y_2-T_E)} \end{pmatrix}. \tag{12}$$

Here, the day of sibling egg-laying, $y_2$, is summed over days $t_i$ through $(t_i + (T_A - 1))$, for consistency with the mother's potential lifespan (Fig 3A). The first term in the summation represents the probability that the mother is alive on the day of sibling egg-laying, and the second term (in larger brackets) represents the expected larval output of the mother on day $t_2$. This latter term is the same as for the mother-larval offspring case, with the exception that the indicator function limits consideration to cases where the day of sibling egg-laying, $y_2$, is between days $(t_2 - T_E - T_L)$ and $(t_2 - T_E)$, for consistency with a larval sibling being sampled on day $t_2$. We provide full-sibling kinship probabilities for other life stage pairs in S1 Text §2.3.

**2.2.4 Half-siblings.** Next, we consider the half-sibling kinship probability for larva-larva pairs, $P_{HSLL}(t_2|t_1)$, which represents the probability that, given a larva sampled on day $t_1$, a larva sampled on day $t_2$ is their half-sibling. This can be expressed as the relative larval reproductive output on day $t_2$ of adult females that mate with the father of a larva sampled on day $t_1$:

$$P_{HSLL}(t_2|t_1) = \frac{\mathbb{E}[\text{Larvae at time } t_2 \text{ that are half-siblings of a larva sampled at time } t_1]}{\mathbb{E}[\text{Larval offspring at time } t_2 \text{ from all adult females}]} = \frac{E_{HSLL}(t_2|t_1)}{E_L}. \tag{13}$$

Here, $E_{HSLL}(t_2|t_1)$ represents the expected number of surviving larvae on day $t_2$ that are half-siblings of a larva sampled on day $t_1$ and is graphically depicted in Fig 3B. $E_L$ is given by Eq 2. For convenience, let us refer to the larva sampled on day $t_1$ as individual 1. To calculate $E_{HSLL}(t_2|t_1)$, there are three unknown event times that we treat as latent variables and take an expectation over—i) the day that egg 1 is laid, $y_1$, ii) the day of the mating event between individual 1's mother and father, $t_i$, and iii) the day that individual 1's father emerges as an adult,

$t_j$:

$$E_{HSLL}(t_2|t_1) = \sum_{y_1=t_1-T_E-(T_L-1)}^{t_1-T_E} p_L(t_1-y_1-T_E) \times \sum_{t_i=y_1-(T_A-1)}^{y_1} p_A(y_1-t_i)$$

$$\times \sum_{t_j=t_i-(T_A-1)}^{t_i} p_A(t_i-t_j) \times E_{HSLL}(t_2|t_1,y_1,t_i,t_j). \tag{14}$$

Here, i) the expectation over the day that egg 1 is laid, $y_1$, is taken over days $(t_1 - T_E - (T_L - 1))$ through $(t_1 - T_E)$, for consistency with the day that larva 1 is sampled, ii) the expectation over the day of the mating event, $t_i$, is taken over days $(y_1 - (T_A - 1))$ through $y_1$, for consistency with egg 1 being laid during their mother's potential lifetime, and iii) the expectation over the day that their father emerges, $t_j$, is taken over days $(t_i - (T_A - 1))$ through $t_i$, so that the mating event overlaps with their father's potential lifetime (Fig 3C). The term $E_{HSLL}(t_2|t_1, y_1, t_i, t_j)$ represents the expected number of surviving larvae on day $t_2$ that are half-siblings of adult 1, conditional upon adult 1 being sampled on day $t_1$, egg 1 being laid on day $y_1$, their mother and father mating on day $t_i$, and their father emerging as an adult on day $t_j$. Additionally, $p_L(t_1 - y_1 - T_E)$ represents the probability that larva 1 has age $(t_1 - y_1 - T_E)$, which is equivalent to the probability that the day of egg-laying was $y_1$, $p_A(y_1 - t_i)$ represents the probability that their mother has age $(y_1 - t_i)$, which is equivalent to the probability that their mother emerged and mated on day $t_i$, and $p_A(t_i - t_j)$ represents the probability that their father has age $(t_i - t_j)$, which is equivalent to the probability that their father emerged on day $t_j$. In general, $p_A(t)$ and $p_L(t)$ are given by Eqs 9 and 13, respectively. $E_{HSLL}(t_2|t_1, y_1, t_i, t_j)$ is then given by:

$$E_{HSLL}(t_2|t_1,y_1,t_i,t_j) = \sum_{t_k=t_j}^{t_j+(T_A-1)} (1-\mu_A)^{(t_k-t_j)} \times \mu_A \times \sum_{y_2=t_k}^{t_k+(T_A-1)} (1-\mu_A)^{(y_2-t_k)} \times \begin{pmatrix} \mathbb{I}[(t_2-T_E-T_L) < y_2 \\ \leq (t_2-T_E)] \\ \times \beta \times (1-\mu_E)^{T_E} \\ \times (1-\mu_L)^{(t_2-y_2-T_E)} \end{pmatrix}. \tag{15}$$

In order to produce a half-sibling, larva 1's father must mate with another adult female and that adult female must produce an offspring. Here, the day of the second mating event, $t_k$, is summed over days $t_j$ through $(t_j + (T_A - 1))$, for consistency with the father's potential lifespan, and the day of sibling egg-laying, $y_2$, is summed over days $t_k$ through $(t_k + (T_A - 1))$, for consistency with the mother's potential lifespan (Fig 3B). The terms in the first summation represent: i) the probability that the father survives days $t_k$ through $t_j$ and therefore is alive on the day of the second mating event, and ii) the probability that the father mates on this day. This latter probability is equal to the adult mortality rate, $\mu_A$, since, for a population at equilibrium, the adult emergence and mortality rates are the same, and the mating rate is equal to the emergence rate since females are assumed to mate upon emergence. Finally, the terms in the second summation represent the probability that the mother is alive on the day of sibling egg-laying, and the expected larval output of the mother on day $t_2$. This latter term (in big brackets) is the same as that for the full-sibling larva-larva case. We provide half-sibling kinship probabilities for other life stage pairs in S1 Text §2.4.

**2.2.5 Numerical calculation.** In Fig 4, selected kinship probabilities are depicted as a function of time between samples, $t_2 - t_1$, for a population of 3,000 adult *Ae. aegypti* with bionomic parameters listed in Table 1. From this, the expected temporal distribution of sampled close-kin is apparent. For mother-larval offspring pairs, for instance, the mother is more likely to be sampled after the larval offspring since, after egg-laying, the mean time to maternal sampling is longer than the duration of the egg stage plus the mean time to larval sampling (Fig

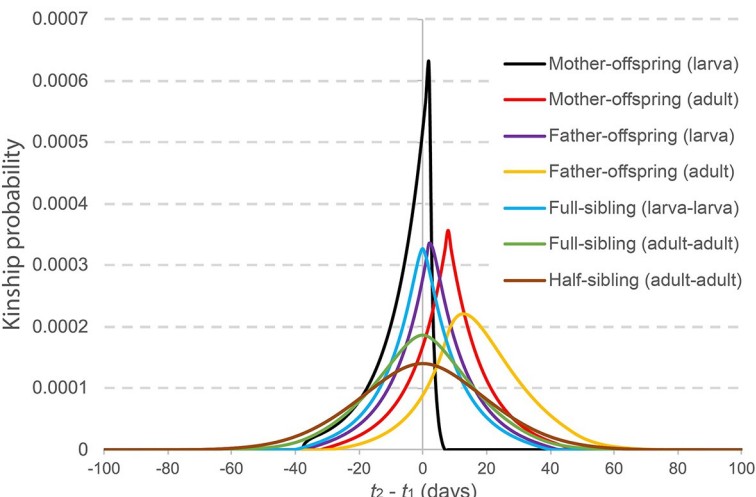

**Fig 4. Kinship probabilities versus time between samples.** Selected kinship probabilities are depicted as a function of time between samples, $t_2 - t_1$. Parent-offspring kinship probabilities are the probability that an individual sampled on day $t_2$ is an offspring of a given adult sampled on day $t_1$. Sibling kinship probabilities are the probability that an individual sampled on day $t_2$ is a sibling of a given individual sampled on day $t_1$. Each probability is calculated as the reproductive output having that relationship divided by the total reproductive output of all adult females in the population, as described in Table 2. The modeled population consists of 3,000 adult *Ae. aegypti* with bionomic parameters listed in Table 1.

2B). Conversely, for father-adult offspring pairs, the father is more likely to be sampled before the adult offspring since, after mating, the mean time to paternal sampling is shorter than the time to maternal egg-laying combined with the duration of offspring juvenile development plus the mean time to adult offspring sampling (Fig 2D). Siblings are most likely to be sampled around the same time as each other, while half-siblings may be found the largest number of days apart (Fig 3).

**2.2.6 Unknown sampling day.** We also consider the case where a sample is collected on day $t$, but the individual could have been sampled (i.e., trapped) on any of the $\tau$ days leading up to and including its collection day. The kinship probability on day $t$ is then the expressed as an expectation over the $\tau$ possible sampling days, each of which is equally likely. We denote the kinship probability in this case with an overline, $\overline{P}(t)$.

As a demonstration, let us consider the mother-larval offspring kinship probability, $\overline{P_{MOL}}(t_2|t_1)$, where the collection days of the adult female and larva are $t_1$ and $t_2$, respectively; but the sampling days of the adult female and larva could be any of the $\tau_1$ and $\tau_2$ days leading up to and including their respective collection days. Any of the potential sampling days of the adult female and larva are equally likely, and so the mother-larval offspring kinship probability may be expressed as the expectation:

$$\overline{P_{MOL}}(t_2|t_1) = \frac{1}{\tau_1 \tau_2} \sum_{i=t_1-(\tau_1-1)}^{t_1} \sum_{j=t_2-(\tau_2-1)}^{t_2} P_{MOL}(j|i). \tag{16}$$

Here, $P_{MOL}(j|i)$ is as defined in Eq 1. Kinship probabilities when sampling days are unknown follow an equivalent formulation for all other kinship categories and sampled sexes and life stages.

### 2.3 Pseudo-likelihood calculation

The goal of this mosquito CKMR analysis is to make inferences about demographic and life history parameters given data on the frequency and timing of observed close-kin pairs. Here, we combine the probabilities of parent-offspring and sibling pairs in a manner that takes advantage of the nature of the kinship probabilities and the sampling process. The joint kinship distribution is too complicated to be expressed analytically as a full likelihood, and so we instead adopt a "pseudo-likelihood" approach [1], which treats the marginal kinship probabilities for each pair of sampled individuals separately. This approach has been shown to produce accurate parameter and variance estimates provided the size of each sample taken is small relative to the overall population [2, 3].

**2.3.1 Parent-offspring pairs.**   Let us begin by considering the mother-larval offspring kinship probability, $P_{MOL}(t_2|t_1)$, which represents the probability that, given an adult female sampled on day $t_1$, a larva sampled on day $t_2$ is her offspring. Now consider $n_F(t_1)$ adult females sampled on day $t_1$. The probability that a given larva has a mother amongst the $n_F(t_1)$ sampled adult females, $p_{MOL}(t_2|t_1)$, is equal to one minus the probability that none of the $n_F(t_1)$ sampled adult females are the larva's mother, i.e.:

$$p_{MOL}(t_2|t_1) = 1 - (1 - P_{MOL}(t_2|t_1))^{n_F(t_1)}. \tag{17}$$

Here, $P_{MOL}(t_2|t_1)$ is as defined in Eq 1. Now consider $n_L(t_2)$ larvae sampled on day $t_2$, and let $k_{MOL}(t_2|t_1)$ be the number of larvae sampled on day $t_2$ that have a mother amongst the adult females sampled on day $t_1$. The pseudo-likelihood that $k_{MOL}(t_2|t_1)$ of the $n_L(t_2)$ larvae sampled on day $t_2$ have a mother amongst the adult females sampled on day $t_1$ follows from the binomial distribution:

$$L(k_{MOL}(t_2|t_1)) = \text{ Binomial } (k_{MOL}(t_2|t_1) : n_L(t_2), p_{MOL}(t_2|t_1)). \tag{18}$$

The full log-pseudo-likelihood for mother-larval offspring pairs, $\Lambda_{MOL}$, follows from summing the log-pseudo-likelihood over all adult female sampling days, $t_1$, and over consistent larval offspring sampling days, $t_2$:

$$\Lambda_{MOL} = \sum_{t_1} \sum_{t_2 = t_1 + T_E - (T_A - 1)}^{t_1 + T_E + (T_L - 1)} \log L(k_{MOL}(t_2|t_1)). \tag{19}$$

Note that, for the purpose of efficient computation, we consider consistent adult sampling days from $(t_1 + T_E - (T_A - 1))$ through $(t_1 + T_E + (T_L - 1))$. The earliest larval sampling day (relative to $t_1$) corresponds to the case where the mother laid the offspring at the beginning of her life, was sampled at the end of her life, and the larval offspring was sampled at the beginning of its life. The latest larval sampling day (relative to $t_1$) corresponds to the case where the mother was sampled on the day they laid their offspring, and the larval offspring was sampled at the end of its life.

Parent-offspring pseudo-likelihood equations for other sampled sexes and life stages follow an equivalent formulation. Of note, consistent offspring sampling days are specific to the kinship and sampled life stages being considered (these can be deduced from schematic diagrams like those in Fig 2). Also, for cases of adult offspring where $t_1 = t_2$, the number of sampled adults, $n_A(t_2)$, is reduced by one to account for the fact that an adult cannot be its own parent.

The joint log-pseudo-likelihood for all parent-offspring pairs is then given by:

$$\Lambda_{PO} = \Lambda_{MOL} + \Lambda_{MOP} + \Lambda_{MOA} + \Lambda_{FOL} + \Lambda_{FOP} + \Lambda_{FOA}. \tag{20}$$

Here, $\Lambda_{MOP}$, $\Lambda_{MOA}$, $\Lambda_{FOL}$, $\Lambda_{FOP}$ and $\Lambda_{FOA}$ denote the log-pseudo-likelihoods for mother-pupal offspring pairs, mother-adult offspring pairs, father-larval offspring pairs, father-pupal offspring pairs and father-adult offspring pairs, respectively.

**2.3.2 Sibling pairs.** For siblings, we adopt a multinomial approach in which each pair of individuals can either be full-siblings, half-siblings or neither. We begin with the larva-larva full-sibling kinship probability, $P_{FSLL}(t_2|t_1)$, defined in Eq 9, and the larva-larva half-sibling kinship probability, $P_{HSLL}(t_2|t_1)$, defined in Eq 13. These represent the probabilities that, given a larva sampled on day $t_1$, a larva sampled on day $t_2$ is their full or half-sibling, respectively. We consider a given larva, indexed by $i$ and sampled on day $t_1(i)$, and $n_L(t_2)$ larvae sampled on day $t_2$, and let $k_{FSLL}(t_2|i)$ and $k_{HSLL}(t_2|i)$ be the number of larvae sampled on day $t_2$ that are full and half-siblings of larva $i$, respectively. When counting siblings, we only consider siblings with indices $> i$ to avoid double-counting. The pseudo-likelihood that $k_{FSLL}(t_2|i)$ and $k_{HSLL}(t_2|i)$ of the $n_L(t_2)$ sampled larvae on day $t_2$ are full and half-siblings of larva $i$, respectively, follows from the multinomial distribution:

$$L(k_{FSLL}(t_2|i), k_{HSLL}(t_2|i)) = \text{Multinomial} \begin{pmatrix} \{k_{FSLL}(t_2|i), k_{HSLL}(t_2|i)\} : \\ n_L(t_2), \{P_{FSLL}(t_2|t_1(i)), P_{HSLL}(t_2|t_1(i))\} \end{pmatrix}. \tag{21}$$

Note that, for cases where $t_1(i) = t_2$, the number of sampled larvae on day $t_2$, $n_L(t_2)$, is reduced by one to account for the fact that a larva cannot be its own sibling. The full log-pseudo-likelihood for larva-larva sibling pairs, $\Lambda_{SLL}$, follows from summing the log-pseudo-likelihood over all sampled larvae, $i$, and over consistent larval sampling days, $t_2$:

$$\Lambda_{SLL} = \sum_{i=1}^{n_L-1} \sum_{t_2 = t_1(i) - 2(T_A-1) - (T_L-1)}^{t_1(i) + 2(T_A-1) + (T_L-1)} \log L(k_{FSLL}(t_2|i), k_{FHLL}(t_2|i)). \tag{22}$$

Consistent larval sampling days for this case are from $(t_1(i) - 2(T_A - 1) - (T_L - 1))$ through $(t_1(i) + 2(T_A - 1) + (T_L - 1))$. The earliest larval sampling day (relative to $t_1(i)$) corresponds to the half-sibling case where the father mated with mother 2 at the beginning of his life and mother 1 at the end of his life. Mother 1 then laid individual 1 at the end of her life, and larva 1 was sampled just before development into a pupa, while mother 2 laid individual 2 at the beginning of her life, and larva 2 was sampled soon after emergence as a larva. The latest larval sampling day (relative to $t_1(i)$) corresponds to the reverse case. Full and half-sibling pseudo-likelihood equations for other life stage pairs follow an equivalent formulation, with consistent sampling days specific to the kinship and sampled life stages being considered (these can be deduced from event history diagrams like those in Fig 3). The joint log-pseudo-likelihood for full and half-sibling pairs is then given by:

$$\Lambda_S = \Lambda_{SLL} + \Lambda_{SLP} + \Lambda_{SLA} + \Lambda_{SPL} + \Lambda_{SPP} + \Lambda_{SPA} + \Lambda_{SAL} + \Lambda_{SAP} + \Lambda_{SAA}. \tag{23}$$

Here, $\Lambda_{SLP}$, $\Lambda_{SLA}$, $\Lambda_{SPL}$, $\Lambda_{SPP}$, $\Lambda_{SPA}$, $\Lambda_{SAL}$, $\Lambda_{SAP}$ and $\Lambda_{SAA}$ denote the log-pseudo-likelihoods for larva-pupa, larva-adult, pupa-larva, pupa-pupa, pupa-adult, adult-larva, adult-pupa and adult-adult full and half-sibling pairs, respectively.

**2.3.3 Parameter inference.** Despite parent-offspring and sibling kinship probabilities not being independent, the pseudo-likelihood approach enables us to combine these pseudo-likelihoods, provided each sample taken is small relative to the overall population [1]. As we will see later, our simulation studies suggest this to be the case. We therefore combine these log-

pseudo-likelihoods to obtain a log-pseudo-likelihood for the entire data set:

$$\Lambda = \Lambda_{PO} + \Lambda_S. \tag{24}$$

Parameter inference can then proceed by varying a subset of the demographic and life history parameters in Table 1 in order to minimize $-\Lambda$. We used the **nlminb** function implemented in the **optimx** function in R [25] to perform our optimizations. This function implements a Newton-type algorithm and performed the best, in terms of speed and accuracy, among the 13 algorithms available through the **optimx** function.

## 2.4 Individual-based simulation model

We developed an individual-based simulation model of mosquito life history to test the effectiveness of the CKMR approach at estimating mosquito demographic and bionomic parameters. The model is an individual-based adaptation of our previous model, **MGDrivE** [26], which is a genetic and spatial extension of the lumped age-class model applied to mosquitoes by Hancock and Godfray [17] and Deredec *et al.* [18] (Fig 1). The simulation time-step is one day. Functionality is included to account for spatial population structure; however, in the present analysis, a single panmictic population is modeled. This population is partitioned according to discrete life stages—egg, larva, pupa and adult—with sub-adult stages having fixed durations as defined earlier. Individual survival through a given day is assumed to follow a Bernoulli distribution with a stage-specific probability. Density-independent juvenile mortality rates are calculated for consistency with observed population growth rates for *Ae. aegypti* (Table 1). Additional density-dependent mortality occurs at the larval stage and regulates population size (see S1 Text §1 for formulae and derivations). Sex is modeled at the adult stage—half of pupae emerge as females, and the other half as males, implemented according to a Bernoulli distribution with probability 0.5. Females mate once upon emergence, with the male mate being chosen at random. Males mate throughout their lifespan, and independently of previous mating events. Females lay eggs at a daily fecundity rate, $\beta$, for the duration of their lifespan with daily egg production of each adult female following a Poisson distribution.

Sampling is lethal, and is implemented as specified, with collection days, locations and sampling rates for each life stage defined by the user. To enable close-kin relationships to be inferred for sampled individuals, each individual is labeled with a unique IN, and parental INs are stored as attributes. Output CSV (comma-separated value) files are produced for each sampled life stage (larva, pupa, adult female and adult male, as appropriate), and include the time (day) and location (patch) of collection, as well as the individual's age at the time of sampling, their IN, and maternal and paternal INs. Inference of mother-offspring, father-offspring, full-sibling and (paternal) half-sibling pairs from this data is straightforward. Age information was not used in this analysis; but may be useful in the future as new technologies emerge to estimate the age of wild-caught adults [27].

## 3 Results

We used simulated data from the individual-based mosquito model to explore the feasibility of CKMR methods to infer demographic and bionomic parameters for *Ae. aegypti*. Our simulated population consisted of 3,000 adults with bionomic parameters listed in Table 1. Open questions concern the suitability of CKMR methods for *Ae. aegypti*, the range of demographic and bionomic parameters that can be accurately estimated using them, and the potential to measure intervention impact. To address these questions, we explored logistically feasible sampling schemes to accurately infer adult and juvenile parameters by varying: i) sampled life stage (larva, pupa or adult) and sex (adult female or male), ii) sampling frequency (daily,

biweekly, weekly or fortnightly), iii) sampling duration (1–4 months), and iv) total sample size (500–5,000 sequenced individuals). For adults, we focused on adult population size, $N_A$, and mortality rate, $\mu_A$, and for juvenile life stages, we focused on larval mortality rate, $\mu_L$, and the duration of the larval stage, $T_L$. By default, our pseudo-likelihood calculations were based on parent-offspring and full-sibling pairs. Half-sibling pairs were only included for optimal sampling schemes, due to the computational burden that half-siblings present by requiring summation over six latent event times (Fig 3B). We also considered subsets of likelihood components in our analyses, in the event that these may provide increased accuracy or precision. Finally, we explored whether only knowing the sampling day to an accuracy of 2–4 days (the number of days between collections) would substantively impact parameter inference.

### 3.1 Optimal sampling schemes to estimate adult parameters

To estimate adult parameters, our default sampling scheme consisted of a total of 1,000 sequenced individuals sampled daily over a three-month period (i.e., ca. 11–12 individuals sampled each day, for a total of 1,000 individuals after three months of sampling). We first explored the optimal distribution of sampled life stage and sex to estimate $N_A$ and $\mu_A$. Sampled larval, adult female and adult male life stage proportions were varied in 25% increments and limited to scenarios where the number of sampled adult females was greater than or equal to the number of sampled adult males (this reflects the case in the field due to the relative difficulty of sampling adult males). We also considered the case where only pupae were sampled, as pupae are often used as indicators of adult population size in entomological field studies [28]. Results of 100 simulation-and-analysis replicates for each of ten sampling scenarios are depicted in Fig 5A and 5B. The key result from this analysis is that the most accurate estimates of $N_A$ and $\mu_A$—in terms of both accuracy of the median and tightness of the interquartile range (IQR)—are obtained when samples are dominated by adult females (75% or higher). This is an intuitive result, as $N_A$ and $\mu_A$ both describe the adult population, and adult females provide the most direct information on kinship—i.e., calculating the kinship probability for father-offspring pairs as compared to mother-offspring pairs involves summing over an additional latent event time (Fig 2). Other key messages from this analysis are that IQRs of inferred parameters are wider for samples dominated by larvae (75% or higher) or pupae (100%), and there is a slight bias towards higher estimates of population size and lower estimates of adult mortality in all cases. Given these results, we focused on adult female sampling while refining other details of the sampling schemes for estimating adult parameters.

Next, we explored the most efficient sampling frequency to estimate $N_A$ and $\mu_A$. While we consider daily sampling a theoretical gold standard, mosquito collections in the field tend to be at most biweekly [29], with weekly collections being more common [7]. For completeness, we also considered collections occurring every two weeks, with results of 100 replicates for each of the four sampling scenarios depicted in Fig 5C and 5D. The key result from this analysis is that CKMR estimates of $N_A$ and $\mu_A$ are robust for daily, biweekly, weekly, and even fortnightly collections, which is reassuring for the logistical feasibility of the method. In the field, the decision on sampling frequency will be based on the required total sample size, and the sampling frequency required to achieve it. We decided to focus on biweekly sampling henceforth, given its precedent in the field, and considering it allows more mosquitoes to be collected than weekly sampling. Here, we assume that the day of sampling is known (i.e., that mosquitoes are collected within a single day of trapping); however, we later relax this assumption so that the recorded day of sampling is pooled over the days between collections.

Following this, we explored the most efficient sampling duration to estimate $N_A$ and $\mu_A$. We explored durations of 1–4 months as, given the short generation time of mosquitoes

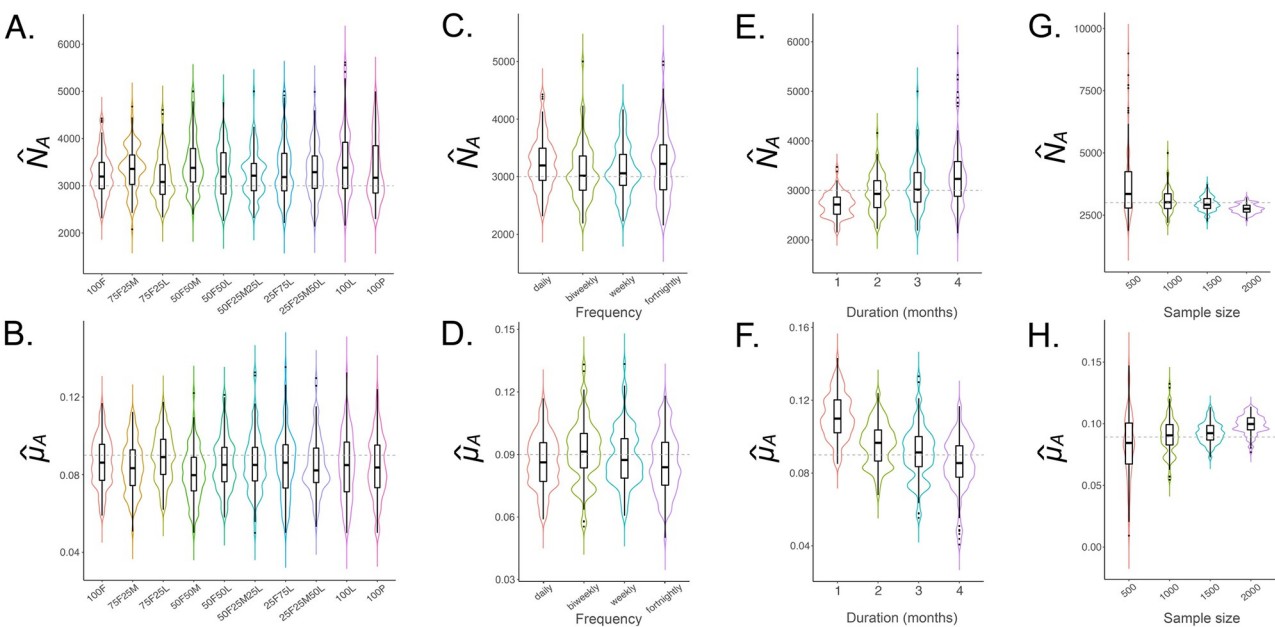

**Fig 5. Sampling schemes to estimate $N_A$ and $\mu_A$ for *Ae. aegypti*.** Violin plots depict estimates of $N_A$ and $\mu_A$ for sampling scenarios described in §3.1. The simulated population consists of 3,000 adult *Ae. aegypti* with bionomic parameters listed in Table 1. Boxes depict median and interquartile ranges of 100 simulation-and-analysis replicates for each scenario, thin lines represent 5% and 95% quantiles, points represent outliers, and kernel density plots are superimposed. The default sampling scheme consists of 1,000 individuals sampled as ca. 11–12 individuals per day over three months. In panels **(A-B)**, sampled larval, adult female and adult male life stage proportions are varied in 25% increments and limited to scenarios where the number of sampled adult females exceeds the number of sampled adult males (e.g., "75F25M" represents a sample consisting of 75% adult females and 25% adult males, and "50F25M25L" represents a sample consisting of 50% adult females, 25% adult males, and 25% larvae). The case of 100% sampled pupae is also included. In panels **(C-D)**, all sampled individuals are adult females, and four sampling frequencies are considered—daily, biweekly, weekly and fortnightly. In panels **(E-F)**, biweekly sampling is adopted, and sampling durations of 1–4 months are explored. In panels **(G-H)**, a sampling duration of three months is adopted, and total sample sizes of 500, 1,000, 1,500 and 2,000 adult females are explored.

[14], parent-offspring pairs could potentially be collected within a month, and given the seasonality of mosquito populations, a maximum sampling period of four months corresponds to a season when the constant population assumption may approximately apply. 100 replicates for each of four sampling scenarios are depicted in Fig 5E and 5F. These results suggest that sampling durations of 3–4 months provide unbiased estimates of $N_A$ and $\mu_A$, while sampling durations of 1–2 months lead to substantively higher estimates of $\mu_A$ and lower estimates of $N_A$. This is at least partly due to compressing the same amount of lethal sampling into shorter time frames resulting in elevated mortality rates and suppressed population sizes. Given these results, we retained a three-month sampling period as the most accurate and efficient option.

Next, we explored the optimal sample size to estimate $N_A$ and $\mu_A$ for *Ae. aegypti*. We performed 100 simulation-and-analysis replicates for each of four total sample sizes—500, 1,000, 1,500 and 2,000 adult females—depicted in Fig 5G and 5H. Results suggest that, while estimates of $N_A$ and $\mu_A$ become more precise for larger sample sizes (as measured by the IQR), adult mortality is notably overestimated for total sample sizes of 1,500 or higher, and adult population size is correspondingly underestimated. This is a reflection of lethal sampling removing individuals from the population and hence increasing adult mortality and reducing adult population size. We therefore converged on an optimal sample size of 1,000 adult females, collected biweekly over a three month period (i.e., ca. 40 adult females per collection), as providing accurate and unbiased estimates of $N_A$ and $\mu_A$.

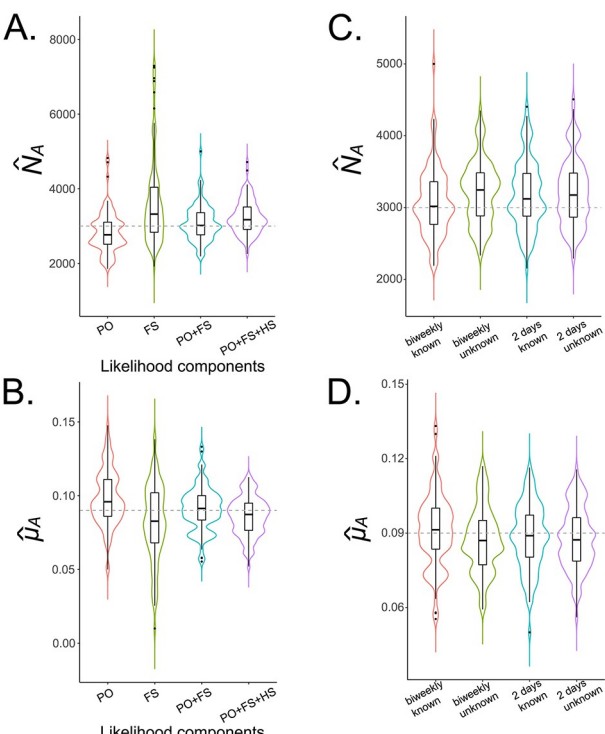

**Fig 6. Pseudo-likelihood components to estimate $N_A$ and $\mu_A$ for *Ae. aegypti*.** Violin plots depict estimates of $N_A$ **(A)** and $\mu_A$ **(B)** for the optimal sampling scheme determined in Fig 4 (1,000 adult females collected biweekly over a three month period, i.e., ca. 40 adult females per collection) and various included pseudo-likelihood components. The simulated population consists of 3,000 adult *Ae. aegypti* with bionomic parameters listed in Table 1. Boxes depict median and interquartile ranges of 100 simulation-and-analysis replicates for each scenario, thin lines represent 5% and 95% quantiles, points represent outliers, and kernel density plots are superimposed. Adult parameter estimates inferred from combined parent-offspring and full-sibling pseudo-likelihood components are more accurate than those inferred from either pseudo-likelihood component in isolation and more accurate than those inferred by inclusion of half-sibling pairs. In panels **(C-D)**, parent-offspring and full-sibling pseudo-likelihood components are used and cases of biweekly sampling and sampling every two days are compared, both where the sampling day is known and where the sampling day is only known within the interval between samples.

Given this optimal sampling scheme to estimate $N_A$ and $\mu_A$, we next explored the pseudo-likelihood components used in these analyses. Curiously, we found that including half-siblings in our analyses increased biases in our parameter estimates, leading to a greater underestimate of $\mu_A$ and overestimate of $N_A$ (Fig 6). This could potentially be due to the half-sibling pseudo-likelihood component requiring a longer sampling period to produce accurate parameter estimates, as half-sibling kinship probabilities require summing over several more latent event times than full-sibling and parent-offspring kinship probabilities. Adult parameter estimates inferred from combined parent-offspring and full-sibling pseudo-likelihood components are more accurate and precise (as measured by the median and IQR of replicate parameter estimates, respectively) compared to those inferred from either pseudo-likelihood component in isolation (Fig 6). This confirms the validity of the optimal sampling scheme inferred in Fig 5, which was inferred for combined parent-offspring and full-sibling pseudo-likelihood components. That produced a population size estimate of 3,016 (IQR: 2,765–3,359), and an adult mortality rate estimate of 0.091 per day (IQR: 0.084–0.100 per day).

Finally, we explored the impact that not knowing the sampling day precisely would have on estimation of $N_A$ and $\mu_A$. We compared cases of biweekly sampling and sampling every two days, both where the sampling day is known and where the sampling day is only known within

the interval between samples (as would be the case for a regular mosquito surveillance program) (Fig 6C and 6D). Encouragingly, we found that parameter estimation is still accurate when the precise sampling day is unknown. The precision of the parameter estimates (as measured by IQR) was not substantively different, and not knowing the precise sampling day led to slight overestimates of $N_A$ and underestimates of $\mu_A$; but only by a small amount—mean estimates of $N_A$ were 3,219 and 3,206 for unknown sampling day (biweekly sampling and sampling every two days, respectively), as compared to 3,076 and 3,121 for known sampling day, and mean estimates of $\mu_A$ were 0.087 per day for unknown sampling day (for both biweekly sampling and sampling every two days), as compared to 0.091 and 0.089 per day for known sampling day (biweekly sampling and sampling every two days, respectively).

## 3.2 Optimal sampling schemes to estimate juvenile parameters

Preliminary explorations of sampling schemes to estimate juvenile parameters suggested this was not possible when including all pseudo-likelihood components. We therefore tested pseudo-likelihood components on a component-by-component basis to see whether some were more informative of juvenile parameters than others. We found that mother-larval offspring pairs provided accurate estimates of larval mortality, $\mu_L$, and that mother-adult offspring pairs provided accurate estimates of the duration of the larval stage, $T_L$. We were not able to estimate pupal parameters ($\mu_P$ or $T_P$), likely due to the brevity of this life stage. Preliminary explorations suggested a sample of 1,000 adult females satisfied the adult requirement for larval parameter estimates, and had already been recommended for estimation of $N_A$ and $\mu_A$. We therefore focused our systematic exploration on the supplemental larval sampling requirement to estimate $\mu_L$ and $T_L$. We estimated these parameters simultaneously using a grid search, varying $T_L$ discretely in the range [1, 10], inferring the value of $\mu_L$ that minimized $-\Lambda$ for each value of $T_L$, and determining the values of $\mu_L$ and $T_L$ that minimized $-\Lambda$ overall.

Our default sampling scheme consisted of a total of 1,000 sequenced adult females and an additional number of larvae sampled daily over a three month period. We first explored the optimal larval sample size to estimate $\mu_L$ and $T_L$. We performed 100 simulation-and-analysis replicates for each of four total larval sample sizes—500, 1,000, 2,000 and 4,000—depicted in Fig 7A and 7B). Results suggest that estimates of $\mu_L$ and $T_L$ are unbiased for larval sample sizes of 1,000 or more, but precision of the estimates, particularly of $\mu_L$ (as measured by the IQR), improves as larval sample size is increased. E.g., for a larval sample size of 1,000, the IQR for $\mu_L$ is 0.461–0.673 per day, while for a larval sample size of 4,000, the IQR is 0.479–0.595 per day (the true value is 0.554 per day, Table 1). We proceeded with a sample size of 4,000 larvae in addition to the 1,000 adult females previously recommended, although we note that a larval sample size as small as 1,000 may be adequate for the case of daily sampling.

Next, we explored the most efficient sampling frequency to estimate $\mu_L$ and $T_L$. As for the adult parameter case, we considered four sampling frequencies—daily, biweekly, weekly and fortnightly—with results of 100 replicates for each scenario depicted in Fig 7C and 7D. The key result from this analysis is that CKMR estimates of $\mu_L$ and $T_L$ are accurate and unbiased for daily and biweekly collections; but that weekly and fortnightly collections are inadequate for estimating $\mu_L$ and less reliable for estimating $T_L$. While this is a more frequent sampling requirement than that for estimating adult parameters, there is a precedent for biweekly collections in the field [29]. Biweekly collections were also our default recommendation for adult collections due to their field precedent, and because they allow a greater number of individuals to be collected over time.

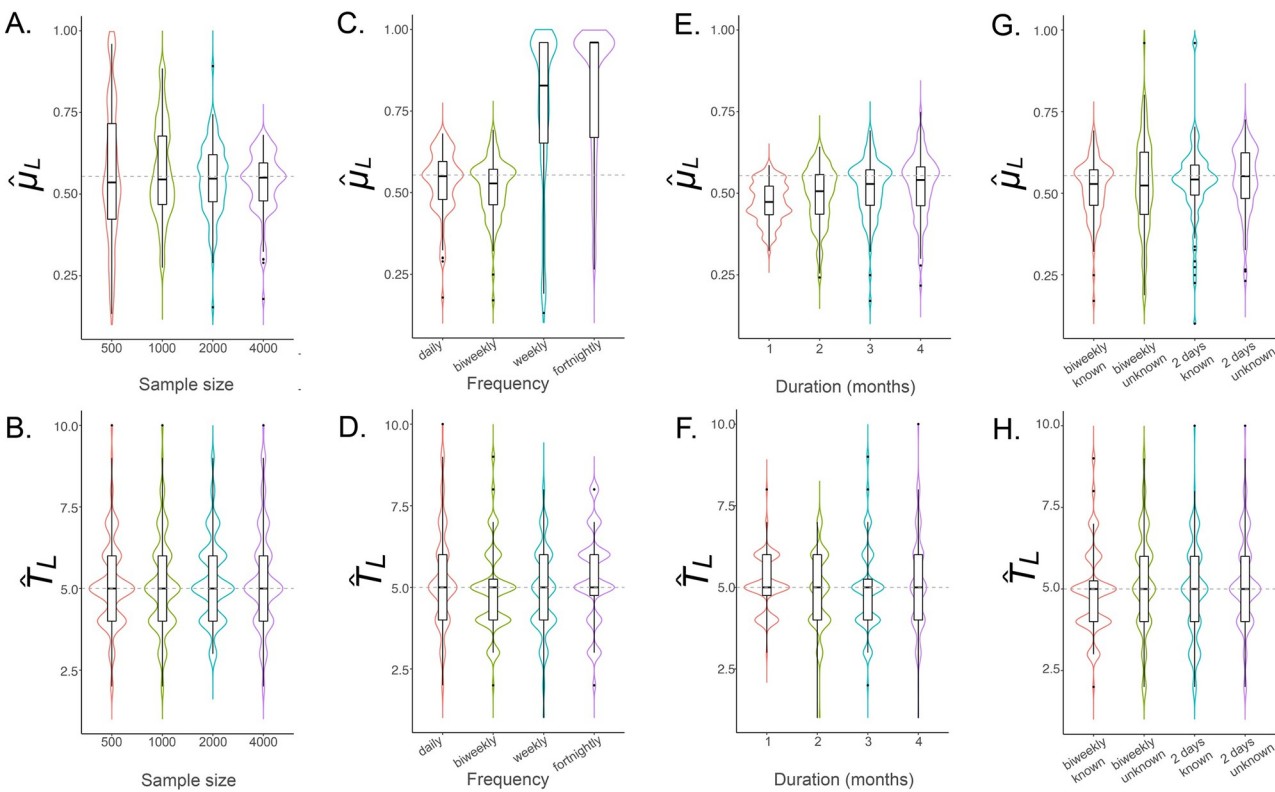

**Fig 7. Sampling schemes to estimate $\mu_L$ and $T_L$ for *Ae. aegypti*.** Violin plots depict estimates of $\mu_L$ and $T_L$ for sampling scenarios described in §3.2. The simulated population consists of 3,000 adult *Ae. aegypti* with bionomic parameters listed in Table 1. Boxes depict median and interquartile ranges of 100 simulation-and-analysis replicates for each scenario, thin lines represent 5% and 95% quantiles, points represent outliers, and kernel density plots are superimposed. The default sampling scheme consists of 1,000 adult females and supplemental larvae sampled daily over a three month period. In panels **(A-B)**, total larval sample sizes of 500, 1,000, 2,000 and 4,000 are explored. In panels **(C-D)**, a larval sample size of 4,000 is adopted, and four sampling frequencies are considered—daily, biweekly, weekly and fortnightly. In panels **(E-F)**, biweekly sampling is adopted, and sampling durations of 1–4 months are explored. The optimal sampling scheme consists of 4,000 larvae and 1,000 adult females collected biweekly over a three month period. In panels **(G-H)**, the optimal sampling scheme is adopted and cases of biweekly sampling and sampling every two days are compared, both where the sampling day is known and where the sampling day is only known within the interval between samples.

We then explored the most efficient sampling duration to estimate $\mu_L$ and $T_L$. As for the adult parameter case, we explored durations of 1–4 months, with results of 100 replicates for each scenario depicted in Fig 7E and 7F. These results suggest that sampling durations of 3–4 months provide accurate estimates of $\mu_L$ and $T_L$, while sampling durations of 1–2 months lead to larval mortality being underestimated, and estimates of $T_L$ being less accurate. We therefore converged on an optimal sample size of 4,000 larvae supplementing the 1,000 adult females recommended earlier, collected biweekly over a three month period, as providing accurate and unbiased estimates of $\mu_L$ and $T_L$. This produces parameter estimates for $\mu_L$ of 0.527 per day (IQR: 0.463–0.571 per day), and for $T_L$ of 5 days (IQR: 4–5 days).

Finally, we explored the impact that not knowing the sampling day precisely would have on estimation of $\mu_L$ and $T_L$. As for adult parameters, we compared the cases of biweekly sampling or sampling every two days where the sampling day is known or only known within the interval between samples (Fig 7G and 7H). Encouragingly, as for adult parameters, we found that parameter estimation was still accurate when the precise sampling day was not known. The precision of $T_L$ estimates was unchanged (as measured by IQR) while the precision of $\mu_L$ estimates decreased slightly when the sampling day was unknown. Not knowing the precise

sampling day led to slight underestimates of $\mu_L$ for biweekly sampling; but only by a small amount—the mean estimate of $\mu_L$ was 0.513 per day for unknown sampling day, as compared to 0.527 per day for known sampling day (as per Table 1, the true value is 0.554 per day).

## 3.3 Measuring intervention impact

Following determination of optimal sampling schemes to infer adult and larval demographic parameters for *Ae. aegypti* populations, we explored whether CKMR methods could be used to infer the impact of an intervention on adult demographic parameters. We considered fogging with insecticides as a commonplace intervention against *Ae. aegypti* that is expected to increase adult mortality rate and decrease population size, and modeled this by increasing the daily lethal sampling rate by increments of 0.01 per mosquito per day, up to a maximum of an additional 0.10 per mosquito per day. Results of this analysis are depicted in Fig 8 and display a clear pattern of increasing estimated adult mortality rate, in line with that simulated, and decreasing estimated population size, reflecting the impact of increased adult mortality on the population. Fig 8C depicts the statistical power of CKMR methods to detect a decrease in $N_A$ or increase in $\mu_A$ as a result of this intervention. Here, a one-tailed test was used given a type I error rate of 5% and 100 simulation-and-analysis replicates for each case. The power to detect a change resulting from an adult mortality rate increased by 0.05 per mosquito per day is $> 90\%$ for both $N_A$ and $\mu_A$, and for an adult mortality rate increased by 0.06 per mosquito per day, the power to detect a change is $>99\%$ for both parameters. This is an encouraging result for the application of CKMR methods to the assessment of intervention impact.

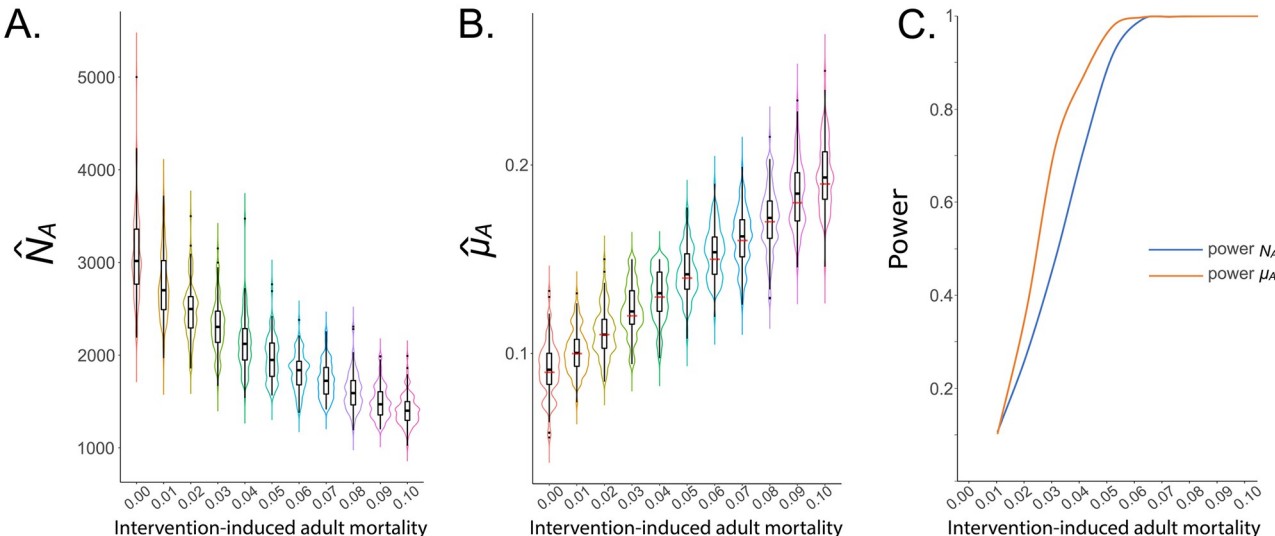

**Fig 8. Application of CKMR methods to infer intervention impact.** CKMR methods are applied to a simulated population of 3,000 adult *Ae. aegypti* with bionomic parameters listed in Table 1 and the optimal sampling scheme for estimating adult parameters (a total of 1,000 adult females sampled biweekly over three months). Fogging is simulated as an intervention that elevates adult mortality. In panels **(A-B)**, violin plots depict estimates of $N_A$ and $\mu_A$ for fogging-induced adult mortality rates of 0.01–0.10 per day (increased in 0.01 per day increments). Boxes depict median and interquartile ranges of 100 simulation-and-analysis replicates for each intervention-induced mortality rate, thin lines represent 5% and 95% quantiles, points represent outliers, and kernel density plots are superimposed. In panel **(B)**, actual intervention-modified adult mortality rates are denoted by red lines. Results depict a pattern of increasing estimated $\mu_A$ and decreasing estimated $N_A$ in response to increasing fogging-induced mortality rates. In panel **(C)**, the statistical power to detect an increase in $\mu_A$ or a decrease in $N_A$ is depicted, assuming a type I error rate of 5%.

## 4 Discussion

We have demonstrated the application of the CKMR formalism described by Bravington *et al*. [1] to estimate demographic parameters for mosquitoes with *Ae. aegypti*, a major vector of dengue, Zika, chikungunya and yellow fever, as a case study. Using an individual-based simulation based on the lumped age-class model [15, 16] applied to mosquitoes [17], we have shown that these methods accurately estimate adult population size, $N_A$, adult mortality rate, $\mu_A$, larval mortality rate, $\mu_L$, and larval life stage duration, $T_L$, for logistically feasible sampling schemes when model assumptions are satisfied. Encouragingly, the optimal sampling scheme inferred from this analysis is compatible with *Ae. aegypti* ecology and field studies. Estimating adult parameters will likely be of most interest, and in this case, only adult females need to be sampled. Conveniently, adult females are preferentially attracted to most commercial traps through cues that mimic potential blood-meals, while adult males are more difficult to trap as they do not blood-feed [29]. Estimating larval parameters requires larval collections, and although larval breeding sites need to be actively sought out, larvae are an abundant life stage that can easily be collected with a cup or pipette [30].

Other details of the CKMR-optimal sampling scheme are also compatible with *Ae. aegypti* ecology. The sampling duration required for accurate estimates of both adult and larval parameters is three months, which is consistent with the length of a season, during which time the constant population size assumption in this analysis approximately holds. For estimating adult parameters, the total sample size of 1,000 adult females collected over three months is reasonable, and sequencing these 1,000 mosquitoes to the extent required to accurately infer close-kin relationships should fall within the budget of current mosquito surveillance programs [7]. For estimating larval parameters, the sample size of 4,000 larvae collected biweekly over three months is achievable, given the abundance of this life stage, although currently the sequencing expense would be burdensome. That said; as sequencing continues to become cheaper, and as more scalable methods become available to estimate relatedness, large-scale larval sequencing may also fall within the budget of surveillance programs, and smaller larval sample sizes are sufficient for daily sampling schemes.

Finally, the sampling frequency requirement of these CKMR methods is compatible with mosquito field studies, with biweekly sampling being adequate for accurate estimation of both adult and larval parameters. This is commonplace among mosquito surveillance programs [29]. If estimates of only adult parameters are desired, sampling frequency can be less frequent (e.g., fortnightly), although achieving the total required sample size may be a barrier to less frequent sampling. For CKMR methods, temporal information contributes to parameter estimation, and so samples will be more informative if the day of collection is known—i.e., if samples from a mosquito trap represent collections for a single day. However, the methods still work adequately if that is not the case—i.e., if samples from a trap represent the accumulation of mosquitoes over several days, as is the case for regular mosquito surveillance programs. A total sample size of 1,000 adult females collected over three months corresponds to biweekly collections of ca. 40 mosquitoes or weekly collections of ca. 80 mosquitoes. With these numbers in mind, the expected daily mosquito yield of a given location can inform the required sampling frequency. An underlying assumption of CKMR analyses is random sampling, which should be noted throughout field protocols. E.g., spatial clustering of samples should be avoided as this can artificially elevate the number of pairs found, and biases towards trapping mosquitoes of certain ages (e.g., those more likely to host-seek) should be avoided where possible, and accounted for where not.

As a preliminary exploration of the application of CKMR methods to mosquitoes, and as a modeling exercise, this study has several limitations. Firstly, the same life history model (Fig 1)

was used as a basis for both the population simulations and the CKMR analysis. Additionally, other than the parameters being estimated, the same parameters were used in both simulations and analysis. This represents an overly generous scenario as compared to the field, where true life history is varied and complex, and where life history parameters are only approximately known. That said; this is an appropriate starting point to verify the utility of the method for mosquitoes—it first needs to be shown to infer the true value of a parameter given the true model. Subsequent analyses should explore the robustness of parameter inference when other parameters in the model are dynamic or misspecified, or when kinship data are generated from a more detailed model. The CIMSiM model of *Ae. aegypti* population dynamics [22], for instance, models juvenile dynamics at the container level and incorporates temperature-driven development rates. Using CKMR methods on data from a detailed model like CIMSiM would allow the impact of structural model differences to be explored to a degree, while acknowledging that true ecological dynamics are more complex than those of any *in silico* model. The impact of modest model variations could also be explored, such as age-dependent mortality rates, variance in the duration of juvenile life stages, and increased variance in the fecundity parameter, *β*. Presently, the daily number of offspring generated by each adult female is Poisson-distributed and distributing this according to an overdispersed negative binomial distribution would reduce the effective population size, $N_e$, while maintaining the census adult population size, $N_A$ [13], the impact of which would be interesting to explore.

A second limitation of the application of our methods is that we have assumed perfect kinship inference throughout. A variety of molecular methods for kinship inference are available [31–33], the accuracy of which should be assessed for *Ae. aegypti* and other species of interest. Incorporating kinship uncertainty into the CKMR likelihood equations is theoretically possible [34], although this has produced little improvement in parameter inference at large computational cost when applied to data from fish species [2]. Likely, the best approach would be to introduce errors in kinship assignment at the simulation phase, and to test the robustness of the methods to this. Here, there is an important distinction between type I (false positive) and II (false negative) error rates. Studies in fish species suggest that kinship inference must have an especially low type I error rate in order for CKMR parameter inference to be informative [1]. Kinship inference methods should be calibrated accordingly. On a related note, there is a debate over the conditions for inclusion of half-siblings in CKMR analyses. Half-sibling relationships are difficult to distinguish from avuncular (e.g., aunt-niece) and grandparent-grandchild relationships, introducing kinship assignment errors into likelihood calculations. Possible solutions have been proposed—e.g., restricting the time window of recording half-sibling pairs to include mostly same-cohort captures [1]—however this is a moot point for the present analysis, given that inclusion of half-siblings reduces the accuracy of parameter estimates even when precisely known.

A third limitation of the current analysis is that it ignores spatial structure. The population of 3,000 adults in the *Ae. aegypti* simulation was based on studies that suggest this to be a reasonable estimate for the number of *Ae. aegypti* adults within a characteristic dispersal radius in a variety of settings [19–21]; however, *Ae. aegypti* adults tend to be relatively sessile, often remaining within the same household unit for the duration of their lifetime [11]. With this in mind, a more accurate model might be *Ae. aegypti* populations distributed across households with migration between them [35]. Areas of future research would be to test the robustness of single-population CKMR methods to data from spatially structured simulations [36], and to incorporate spatial structure into the CKMR analyses themselves, opening the potential to estimate dispersal parameters using these methods. The theoretical underpinnings of this latter approach have been outlined by Bravington *et al.* [1], and an analogous approach limited to discrete generations and parentage data has been used to estimate dispersal parameters for

coral trout [37]. Alternative close-kin methods have also been used to characterize dispersal distances for *Ae. aegypti* [6, 7], and it will be interesting to see whether a spatially structured CKMR approach can infer complementary information.

The application of CKMR methods beyond fish species has been contemplated since their inception [1], and extending their application to the egg-larva-pupa-adult life history of *Ae. aegypti* mosquitoes is promising for their application to insect species with comparable life histories. A species of particular interest is *Anopheles gambiae*, the main African malaria vector, which has a similar life history, increased dispersal [11] and larger population sizes than *Ae. aegypti* [38, 39]. Age-grading methods are also available for this species, based on ovariole measurements and emerging biochemical and spectroscopic techniques [27]. Incorporating approximate age-at-capture information with kinship data should greatly enhance the precision of CKMR parameter inference, as has been seen for applications to southern bluefin tuna [2] and sharks [3]. The larger size of *An. gambiae* populations also means that smaller population proportions need to be sampled in order to obtain accurate parameter estimates [13]. Although the total required sample size will be higher, lethal sampling is less likely to bias the mortality rate estimate upwards and the population size estimate downwards (as seen for *Ae. aegypti* in Fig 5). Several species of insect agricultural crop pests should also be suited to these CKMR methods, including the medfly and spotted wing *Drosophila*; although theoretical assessments will first be needed, especially for more long-lived pest species such as the pink bollworm.

## 5 Conclusions

We have theoretically demonstrated the application of CKMR methods to estimate adult and larval parameters for mosquitoes, with *Ae. aegypti* as a case study. CKMR methods have advantages over traditional mark-release methods, as the mark is genetic, removing the need for physical marking and recapturing. Particularly encouraging is the fact that the inferred optimal sampling scheme is compatible with *Ae. aegypti* ecology and field studies, meaning that the requisite samples may be obtained with only minor adjustments to current mosquito surveillance programs. The methods also appear effective at detecting intervention-induced changes in adult parameters. Sequencing requirements are significant, particularly for estimating larval parameters; however, as sequencing becomes cheaper and more efficient, this will become less burdensome and perhaps even routine. Work remains to test the robustness of these methods under a range of scenarios in which model components and parameters vary, and in which kinship inference is imperfect; however this study represents an important first demonstration that parameter inference is accurate when the underlying model is known. Application to other insects of epidemiological and agricultural significance is promising, particularly for *An. gambiae*, a major malaria vector for which age-grading methods are available.

## Supporting information

**S1 Text. Supplemental model equations.** Additional equations describing the lumped age-class model of mosquito population dynamics, and kinship probabilities for parent-offspring and sibling pairs that, for brevity, were not included in the manuscript.
(PDF)

## Acknowledgments

We thank Dr. Igor Filipović for help with parallelizing code and running simulation replicates, Dr. Eileen Jeffrey Gutiérrez and Dr. Tomás León for discussions regarding mosquito sampling and life history, and Yi Li for help with formulating the kinship probabilities.

## Author Contributions

**Conceptualization:** Gordana Rašić, John M. Marshall.

**Data curation:** Yogita Sharma, Jared B. Bennett.

**Formal analysis:** Yogita Sharma, Jared B. Bennett, John M. Marshall.

**Funding acquisition:** Gordana Rašić, John M. Marshall.

**Investigation:** Yogita Sharma, John M. Marshall.

**Methodology:** Yogita Sharma, Jared B. Bennett, John M. Marshall.

**Project administration:** John M. Marshall.

**Resources:** Gordana Rašić, John M. Marshall.

**Software:** Jared B. Bennett, John M. Marshall.

**Supervision:** John M. Marshall.

**Validation:** John M. Marshall.

**Visualization:** Gordana Rašić, John M. Marshall.

**Writing – original draft:** John M. Marshall.

**Writing – review & editing:** Yogita Sharma, Jared B. Bennett, Gordana Rašić, John M. Marshall.

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
