## [Decision Letter · Decision Letter 0]

9 May 2022

Dear Marshall,

Thank you very much for submitting your manuscript "Close-kin mark-recapture methods to estimate demographic parameters of mosquitoes" for consideration at PLOS Computational Biology.

As with all papers reviewed by the journal, your manuscript was reviewed by members of the editorial board and by several independent reviewers. In light of the reviews (below this email), we would like to invite the resubmission of a significantly-revised version that takes into account the reviewers' comments.

Both reviewers recognise the value of this work, which addresses an important question relating to the paucity of demographic data describing mosquito populations, and uses an interesting computational approach to estimate demographic rates from close kin mark recapture experiments. Reviewer 1 recommends some substantial changes and reviewer 2 raises some important caveats to the analysis that need to be emphasised in the written text of the manuscript.

I agree with Reviewer 1 that the readability of the methods section would benefit from a rewrite to capture the equations conceptually in the main text (perhaps some illustrative plots would be helpful) and designate repetitive material to an appendix. I also agree that the performance of the method when the population is not in equilibrium (growing, declining or fluctuating seasonaly) needs to be clarified in order to provide a balanced assessment, and I'd be particularly interested to know how this influences estimation of daily larval mortality.

We cannot make any decision about publication until we have seen the revised manuscript and your response to the reviewers' comments. Your revised manuscript is also likely to be sent to reviewers for further evaluation.

Sincerely,

Penny A Hancock

Guest Editor

PLOS Computational Biology

Ville Mustonen

Deputy Editor

PLOS Computational Biology

Both reviewers recognise the value of this work, which addresses an important question relating to the paucity of demographic data describing mosquito populations, and uses an interesting computational approach to estimate demographic rates from close kin mark recapture experiments. Reviewer 1 recommends some substantial changes and reviewer 2 raises some important caveats to the analysis that need to be emphasised in the written text of the manuscript.

I agree with Reviewer 1 that the readability of the methods section would benefit from a rewrite to capture the equations conceptually in the main text (perhaps some illustrative plots would be helpful) and designate repetitive material to an appendix. I also agree that the performance of the method when the population is not in equilibrium (growing, declining or fluctuating seasonaly) needs to be clarified in order to provide a balanced assessment, and I'd be particularly interested to know how this influences estimation of daily larval mortality.

Reviewer's Responses to Questions

**Comments to the Authors:**

Reviewer #1: See attachment

Reviewer #2: This is a great paper. It took me longer than I expected to review it, because the analyses are quite involved, but this is unavoidable (both the complexity and the delay). The importance of the question justifies the intensity of the analyses. I support its publication in its current form because I think the analyses are novel and sophisticated, and because the application is internationally important and impactful.

I’ve got one major criticism of the work, and one minor. The central problem is acknowledged by the authors – that it’s pretty easy to fit a model to matching synthetic data. To my mind, model structural error is a big deal – almost big enough to cast doubt on all the results shown here. The problem is that there’s no solution to the problem. All you can do is make the case quite emphatically. Thus, I’d recommend that the authors spend a few more sentences talking about this issue.

My second, and more minor problem, is the reliance on comparisons with marine fish studies. This might be a problem, given how different fish and mosquitoes are. It’s fine to adopt methods from different contexts - dispersal is dispersal, and kinship is kinship. However, type 1 and 2 error rates resulting from the application and analysis of SNP may differ. I’d be really careful what you extrapolate from one context to this one.

**Have the authors made all data and (if applicable) computational code underlying the findings in their manuscript fully available?**

Reviewer #1: None

Reviewer #2: Yes

PLOS authors have the option to publish the peer review history of their article (what does this mean?). If published, this will include your full peer review and any attached files.

Reviewer #1: **Yes: **Ben Lambert

Reviewer #2: No
---

## [Decision Letter · Decision Letter 1]

21 Oct 2022

Dear Marshall,

Thank you very much for submitting your manuscript "Close-kin mark-recapture methods to estimate demographic parameters of mosquitoes" for consideration at PLOS Computational Biology. As with all papers reviewed by the journal, your manuscript was reviewed by members of the editorial board and by several independent reviewers. The reviewers appreciated the attention to an important topic. Based on the reviews, we are likely to accept this manuscript for publication, providing that you modify the manuscript according to the review recommendations.

The reviewers appreciate that most of their suggestions have been addressed. Reviewer 1 has made further suggestions to improve the readability and clarity of the manuscript.

Sincerely,

Penny A Hancock

Guest Editor

PLOS Computational Biology

Ville Mustonen

Section Editor

PLOS Computational Biology

Reviewer's Responses to Questions

**Comments to the Authors:**

Reviewer #1: Uploaded as attachment

**Have the authors made all data and (if applicable) computational code underlying the findings in their manuscript fully available?**

Reviewer #1: None

PLOS authors have the option to publish the peer review history of their article (what does this mean?). If published, this will include your full peer review and any attached files.

Reviewer #1: **Yes: **Ben Lambert

Figure Files:

Data Requirements:

Reproducibility:

References:

---

## [Editor Report · Decision Letter 2]

22 Nov 2022

Dear Marshall,

We are pleased to inform you that your manuscript 'Close-kin mark-recapture methods to estimate demographic parameters of mosquitoes' has been provisionally accepted for publication in PLOS Computational Biology.

Best regards,

Penny A Hancock

Guest Editor

PLOS Computational Biology

Ville Mustonen

Section Editor

PLOS Computational Biology

---

## [Editor Report · Acceptance letter]

7 Dec 2022

PCOMPBIOL-D-22-00257R2 

Close-kin mark-recapture methods to estimate demographic parameters of mosquitoes

Dear Dr Marshall,

I am pleased to inform you that your manuscript has been formally accepted for publication in PLOS Computational Biology. Your manuscript is now with our production department and you will be notified of the publication date in due course.

With kind regards,

Anita Estes
